Consumer-priced wearable sensors combined with deep learning can be used to accurately predict ground reaction forces during various treadmill running conditions

Carter Josh 1 jc2369@bath.ac.uk
Chen Xi 2
http://orcid.org/0000-0001-7877-6755 Cazzola Dario 1
http://orcid.org/0000-0002-9021-8956 Trewartha Grant 3
http://orcid.org/0000-0001-5383-7072 Preatoni Ezio 1
1 Department of Health, University of Bath , Bath, Somerset , United Kingdom
2 Department of Computer Science, University of Bath , Bath, Somerset , United Kingdom
3 School of Health and Life Sciences, University of Teesside , Middlesbrough, North Yorkshire , United Kingdom
Young Jesse
Electronic publication date: 2024 Aug 29
Publication date: 2024
Volume: 12
Electronic Location ID: e17896
Received 2024 Apr 16; Accepted 2024 Jul 19
Copyright: © 2024 Carter et al.
Copyright year: 2024
Copyright holder: Carter et al.
License: This is an open access article distributed under the terms of the Creative Commons Attribution License, which permits unrestricted use, distribution, reproduction and adaptation in any medium and for any purpose provided that it is properly attributed. For attribution, the original author(s), title, publication source (PeerJ) and either DOI or URL of the article must be cited.
License URL: https://creativecommons.org/licenses/by/4.0/

Keywords: Biomechanics, Distance running, Machine learning, Pressure insole, IMU, LSTM, Human locomotion, Training load, Biofeedback

Funding: Nurvv Ltd University of Bath The PhD within which this research was completed was funded by Nurvv Ltd and the University of Bath. Nurvv Ltd has since been dissolved. The funders had no role in study design, data collection and analysis, decision to publish, or preparation of the manuscript.

==============================
Ground reaction force (GRF) data is often collected for the biomechanical analysis of running, due to the performance and injury risk insights that GRF analysis can provide. Traditional methods typically limit GRF collection to controlled lab environments, recent studies have looked to combine the ease of use of wearable sensors with the statistical power of machine learning to estimate continuous GRF data outside of these restrictions. Before such systems can be deployed with confidence outside of the lab they must be shown to be a valid and accurate tool for a wide range of users. The aim of this study was to evaluate how accurately a consumer-priced sensor system could estimate GRFs whilst a heterogeneous group of runners completed a treadmill protocol with three different personalised running speeds and three gradients. Fifty runners (25 female, 25 male) wearing pressure insoles made up of 16 resistive sensors and an inertial measurement unit ran at various speeds and gradients on an instrumented treadmill. A long short term memory (LSTM) neural network was trained to estimate both vertical (GRFv) and anteroposterior (GRFap) force traces using leave one subject out validation. The average relative root mean squared error (rRMSE) was 3.2% and 3.1%, respectively. The mean (GRFv) rRMSE across the evaluated participants ranged from 0.8% to 8.8% and from 1.3% to 17.3% in the (GRFap) estimation. The findings from this study suggest that current consumer-priced sensors could be used to accurately estimate two-dimensional GRFs for a wide range of runners at a variety of running intensities. The estimated kinetics could be used to provide runners with individualised feedback as well as form the basis of data collection for running injury risk factor studies on a much larger scale than is currently possible with lab based methods.

Introduction

A key feature of distance running is repeated contact between the foot and floor, with a ground reaction force (GRF) applied to the body at each step. The combined knowledge of the body’s position and the external force being applied to it is necessary to understand the full biomechanics of the system and to derive downstream biomechanical metrics. Typical gait analysis studies aim to measure the runner’s kinematics and kinetics using lab based equipment, such as video based motion capture and force plates. While these methods provide gold standard levels of accuracy, their collection is often coupled with inherent limitations. The utilisation of expensive equipment requires trained operators, which limits the accessibility of the analysis to the wider running community. Those that are able to undergo lab-based biomechanical gait analysis are often evaluated irregularly, and for short periods of fixed speed treadmill running, which is not reflective of their typical distance running training and the natural variation in speed and route.

To overcome some of these limitations, the use of wearable sensors as a data collection solution for gait analysis has been expanding year on year (Hutabarat, Owaki & Hayashibe, 2021). Many studies have focused on extracting discrete biomechanical gait features from the wearable sensor data (Benson et al., 2018), but more recently several studies have estimated the full time series signals of gait kinematics and kinetics (Gurchiek, Cheney & McGinnis, 2019). Whilst some studies have adopted an estimation approach based on resolving physical laws (Matijevich et al., 2020), many others have opted for data driven estimation methods (Xiang et al., 2022). These data driven approaches rely on the fact that underlying associations exist between the data that is collected from a wearable sensor and data collected from a lab based system. With enough data collected from a lab based and a wearable system in parallel, the lab data can be used to train a model for the estimation of gait kinematics and kinetics from solely wearable data.

A considerable area of focus within running gait research has been on the possible associations between an individual’s running kinetics and their future risk of overload injury (Ceyssens et al., 2019). Whilst the strength of the direct association between GRFs and tibial bone load is still debated (Matijevich et al., 2019), knowledge of GRFs are necessary to calculate further load measures such as net joint moments and joint contact forces. If wearable sensor systems could be used to estimate GRFs during everyday running, a much larger and externally valid dataset could be created. This more consistently occurring biomechanical analysis could then be paired with regular pain and injury assessment, to perform risk factor studies on a much larger scale than currently possible with lab based biomechanical assessment. Before this goal can be realised two barriers must be overcome: the system being used to estimate the biomechanical quantities must be proved to be sufficiently accurate during a variety of running conditions and also deemed accessible and acceptable to many runners.

Recently, researchers have focused on assessing the accuracy with which a machine learning model can estimate GRFs from wearable sensor data during running within a controlled lab environment. Honert et al. (2022) combined pressure insole data with a type of recurrent neural network (RNN), called a long short term memory (LSTM) neural network, to estimate average vertical ground reaction force ( GRFv) curves. The authors reported a relative root mean squared error (rRMSE; the RMSE normalised to the range of the signal) of 4.2% across the stance phase during treadmill running at various speeds and gradients. Similarly, Alcantara et al. (2021) used data from an accelerometer sensor on the sacrum and accelerometers on each shoe to estimate the GRFv of 19 runners at a variety of treadmill speeds and gradients using an LSTM model, achieving an average rRMSE of 6.4 ± 1.5%. Johnson et al. (2021) used motion capture data to construct signals from five ‘virtual IMU’ locations, before evaluating the performance of a convolutional neural network (CNN) for the estimation of ground reaction forces during overground running and cutting movements. During ‘moderate speed’ running an average rRMSE of 13.9% was reported. Whilst these studies have shown the very promising reality that wearable systems can be used to accurately estimate GRFs, many of the studies in this area have typically evaluated performance on smaller subsets of homogeneous runners during a reasonably constrained set of running conditions. Within these studies the sensors being used were often research grade systems, these systems are not designed to be used outside of a lab-based environment where factors such as battery life, durability, and ease of use become important considerations. For this reason these systems can not be readily used by consumers and the barrier for runners to start gaining biomechanical insights from them are high.

Therefore, the aim of this study was to evaluate how accurately a machine learning model could estimate GRFs during varied treadmill running, with wearable data collected from a consumer-focused system (Nurvv Run, Nurvv Ltd, London, UK). The intention during study design was to collect a treadmill running dataset with a greater variety of conditions and amongst a much more heterogeneous population, both in terms of anthropometrics and running experience, than previous studies in this area. Such a dataset would then allow for a more rigorous evaluation of how effective the system could be at estimating kinetics during everyday training amongst a varied runner population.

Methods

Data collection

Fifty distance runners (25 male and 25 female) were recruited to participate in this study if they typically completed at least one 30+ min run per week and were currently free of injury. The intention during participant recruitment was to include a representative pool of runners with a wide variety of anthropometric characteristics and performance capabilities (Table 1). All participants provided written informed consent and all experimental procedures were approved by the University of Bath’s Research Ethics Approval Committee for Health (EP20/21-069).

Table 1 Anthropometric and running-based characteristics of the participants.

All data is shown in the format: ‘mean (standard deviation), min-max’, with the exception of footstrike type showing the percentage of participants defined as forefoot, midfoot, or rearfoot strikers. The bold value indicates the mean value of that metric.

Sex	Age (years)	Mass (kg)	Typical weekly distance (km)	Easy running speed (km/h)	Footstrike distribution (% Fore, Mid, Rear)	
Male	35 (14) 18–69	71.7 (10.3) 52.6–97.0	29 (20) 5–75	11.4 (1.6) 7.5–15.3	20%, 32%, 48%	
Female	28 (10) 18–54	60.1 (7.0) 51.2–75.8	37 (27) 7–100	10.6 (1.4) 8.0–13.3	12%, 28%, 60%	

Before the start of data collection a pair of Nurvv Run insoles of the appropriate size was placed into the participant’s typical running shoe, placed underneath the shoes’ normal insole if possible. The Nurvv Run system (consumer price: $250 USD) is made up of a pair of thin pressure insoles that each contain 16 force sensitive resistors (FSRs) spread evenly across the foot and an attached device that clips on to the lateral aspect of the shoe. The device connected to the left insole contains an inertial measurement unit (IMU). During data collection the Nurvv system collected the FSR output value from all 32 sensors at the frequency of 1,000 Hz. Due to storage limitations on the Nurvv system these values were only logged at 50 Hz (20 ms moving average windows) between the identified initial contact and toe off times of each ground contact, values outside of these events were deleted. The IMU built into the device clipped to the lateral aspect of the left shoe collected and logged data continuously at 1,138 Hz (range of ±4,000 degrees per second for gyroscope and ±30 g for accelerometer). The footstrike classifications used within this article were exported directly from the Nurvv Run system, which uses a custom algorithm that compares the timing of pressure increases under each zone of the foot upon initial contact, informed by the results of Giandolini et al. (2014). The participants completed a running protocol on an instrumented treadmill that collected three-dimensional GRFs at 1,000 Hz (Bertec, OH, USA). The protocol was made up of a flat (+1%, chosen to reflect energetic cost of outdoor running (Jones & Doust, 1996)), uphill (+6%), and downhill (−4%) stage and the running speeds were almost all based around each participant’s typical easy training pace. During the flat condition participants completed 5 min at their easy training speed, 2 min 10% slower and 2 min at 10% faster than this speed. Participants also completed 3 min at the fixed speed of 12 km/h during the flat condition regardless of their chosen speed (three participants were not able to complete this speed). During the uphill condition, 2 min at their chosen speed and 2 min at 10% below their chosen speed was completed. During the downhill condition 2 min was completed at their chosen speed, as well as at 10% above and 10% below this speed.

Data processing

A fourth order low pass Butterworth filter (15 Hz cut off) was applied to the treadmill force data before a 40 N threshold was used to identify the timing of all foot touchdown and toe off events. Before any further processing of the pressure data was completed the logged 50 Hz FSR data was up-sampled back to the recorded frequency of 1,000 Hz using a quadratic interpolator in the Python package Scipy (v1.7.3). The pressure data (FSR signals) were then segmented into contact periods using an internal Nurvv algorithm, which uses the participant’s mass to calculate a total FSR output threshold that is then used to identify the same foot contact events. The same event timings were identified in the IMU data using an accelerometer-based step detection algorithm previously validated for treadmill running (Benson et al., 2019). As all three systems were running on separate clocks, the segmented foot contact data had to be synchronised to ensure the same physical foot contact was being evaluated by each independent system. This was done by correlating the stride times from two systems against each other. The alignment of these two vectors were then adjusted to find the strongest correlation between the sets of stride times calculated from each system, indicative of the correct alignment (Fig. 1). The alignment was accepted if the Pearson correlation coefficient was over 0.85. However, the correct alignment was often very clear ( r>0.99) in comparison to the incorrect alignment (r<0.50) occurring one contact before and after. This process was performed once with stride times from the force data and the pressure data, before then being repeated with the force data and IMU data, ensuring foot contacts from all three systems were synchronised.

Figure 1 Illustration of the correlation approach used to ensure the correct step alignment between the three independent systems.

Coloured numbers above each figure represent the contact number recorded on each of the independent systems. This same process was also repeated to align steps from the force data and IMU data.

Each isolated foot contact had a varying number of data points due to the duration of the ground contact, across all participants the longest contact was 361 ms. Therefore, we normalised all signals to the length of 400 data points, to ensure no resolution was lost from the time series signals. The treadmill force data and the foot IMU data were filtered with fourth order low pass Butterworth filters with a 15 Hz and a 10 Hz cut off, respectively. As the pressure data was logged at 50 Hz (using 20 ms window averaging) before it was then interpolated back to 1,000 Hz, following visual inspection it was deemed that no filtering of the FSR time series data was needed. Based on the pressure data from each of the 16 FSR channels under the left foot, a two-dimensional centre of pressure coordinate ( CoPx, CoPy) throughout each foot contact was calculated using a weighted average approach and the known geometry of the sensors within the insole.

Following the preparation of the time series data at each foot contact, a data matrix had to be created that would serve as the input to the deep learning model for the estimation of the GRF traces. These input features should include as much information about the foot contact as possible whilst remaining feasible to collect continuously during an individual’s running session. The segmented and normalised time series data from the 16 FSR pressure channels, the two calculated centre of pressure values ( CoPx, CoPy), and the three axes of the IMU accelerometer were combined with the following discrete characteristics; duration of ground contact, participant mass, current running speed, current gradient/slope, insole length. All of these input features were chosen due to their feasibility to be collected outside the lab. The discrete values were duplicated for the length of the time sequence [1×400]. Each of these features were then concatenated together into a single input matrix of shape [N×400×26] where N is the number of foot contacts included. Before this input matrix was passed to the model, the data was first standardised to z-scores. This ensures that features with greater magnitudes did not inherently have greater influence on the model’s output, as well as improving numerical stability and convergence speed during model training. The training set was first transformed to z-score values based on each features mean and standard deviation values, then the same transformation was performed to the validation set using the descriptive statistics calculated from the training set. This avoids the ‘leaking’ of any information from the validation set into the training set, replicating the pre-processing conditions upon model implementation.

Machine learning models

Independent deep learning models were created to estimate GRFv and anteroposterior ground reaction forces ( GRFap) from the input matrix in the Python package Pytorch (v1.10) (Paszke et al., 2019), but the same architecture was used to create each model. The matrix of input data was firstly passed to a bi-directional LSTM layer (Hochreiter & Schmidhuber, 1997), which output two hidden state vectors (one for each direction) at each time point in the sequence. These hidden state vectors are the same size as the hidden size of the LSTM network and contain temporal information about the input sequence from the forward and backward pass of the LSTM. The hidden state value at each time point was then passed through three linear layers, transforming the hidden state vector at that time point into an estimated force value (Fig. 2). This is repeated for the LSTM output at every timepoint in the sequence until a full time series of estimated force values are produced. This estimated force trace that the deep learning model produces was then compared to the measured force trace from the treadmill, the PyTorch autograd engine can then leverage its gradient tracking functionality to back propagate through the LSTM model, identifying how the model weights can be appropriately adjusted to decrease the difference between the measured and estimated force trace. The direction and strength of these model weight updates were informed by an Adaptive Moment Estimation (ADAM) optimiser (Kingma & Ba, 2014), aiming to minimise the RMSE between the measured forces and estimated forces across the training batch.

Figure 2 The general architecture of the GRF estimation models.

The model is made up of the two-dimensional time series input matrix for each foot contact, being passed through the bi-directional LSTM layer, after which the hidden state vectors from each time point are passed through three linear layers to map the hidden state vectors to a single predicted force value. The size of the layers and activation functions being used were identified through hyperparameter optimisation, results shown in Table 2.

Table 2 Hyperparameter optimisation results, search spaces were given as either a set of possible hyperparameter input options or a lower and upper bound from which to select a value between.

Hyperparameter	Search space	GRFv model selection	GRFap model selection	
Learning rate	1×10−1→1×10−5	5×10−4	2×10−5	
Number of epochs	1→15	1	8	
Batch size	[8,16,32,64,128]	8	8	
LSTM hidden size	[32,64,128,256,512]	512	512	
Input data dropout rate	0.1→0.5	0.2	0.5	
Linear layer dropout rate	0.1→0.5	0.2	0.4	
Linear layer 2 size	[32,64,128,256,512]	64	512	
Linear layer 3 size	[32,64,128,256,512]	32	512	
Activation function	ReLU, leaky_ReLU, tanh, sigmoid	ReLU	Leaky_ReLU	

To increase the objectivity of the model creation process we chose to use hyperparameter optimisation to identify which hyperparameter choices, used during initial model definition, led to the highest GRF estimation accuracy. A search window for each hyperparameter was chosen and the Python package Optuna (v3.2, Tree-structured Parzen Estimator) was used to find the combination of hyperparameter options that maximised model accuracy. Running a hyperparameter optimisation on the full dataset can lead to the overfitting of hyperparameters to the current dataset, serving to inflate the model’s performance on the dataset within this study but reducing its ability to perform well on unseen data. To evaluate for the presence of this effect, a representative 25 of the 50 participants were separated out based on their anthropometric and running characteristics and split into five groups of five. The Optuna optimiser aimed to select the hyperparameters that minimised the average validation loss across the five folds, set to 250 iterations as a compromise between computational time and exploration of solution space. Within the presentation of the results the 25 participants included in the hyperparameter optimisation will be separated from those who were not, so the presence of hyperparameter overfitting can be evaluated for. After the final hyperparameters had been identified by the optimiser for the GRFv and GRFap model they were used to construct the final model architecture. To ensure an accurate evaluation of the model’s performance on unseen data all data belonging to an individual remained contained in either the training or validation set at any given time. Whilst more computationally intense, Leave One Subject Out (LOSO) validation was performed. This evaluation technique keeps the data for all but one participants within the training set before evaluating the trained model’s performance on the individual’s data left out of training, repeated as many times as there are participants. In comparison to cross fold evaluation, this method provides greater insight into the current model limitations, setting expectations for future use of the trained model. To evaluate how accurately the model was able to estimate force traces at a range of magnitudes a Bland Altman analysis was performed.

To gain a greater understanding of which variables being input into the model were having the largest influence on the accuracy of the model output, Permutation Feature Importance (PFI) was also calculated. The PFI value is calculated by replacing the time series data of a given input variable with a sequence of random numbers, before passing the input matrix to the model (Molnar, 2020). The differing estimated force traces can then be compared, with the PFI value being the ratio between the RMSE after permuted input and the original RMSE value. This process was repeated 100 times for each variable in the input set to minimise the influence of random input values. Larger PFI values indicate a variable having a greater positive influence on the model’s force estimation accuracy, with values under 1.0 indicating the variable having a negative influence on the model’s accuracy.

Results

The hyperparameter optimisation was run for both the GRFv estimation model and the GRFap estimation model. Following all iterations, the hyperparameter combination that lead to the lowest average RMSE across the cross fold validation was selected (Table 2) and taken forward to the full LOSO validation.

Vertical ground reaction force estimation

Across all 50 participants the average RMSE was 0.15 bodyweight (BW) (Fig. 3) and the average correlation between measured and estimated GRFv traces was 0.991. The accuracy with which the model could estimate the GRFv trace varied considerably between participants. For the participant on which the model performed best, the average GRFv estimation error across all foot contacts was 0.05 BW, with an average correlation of 0.999. In comparison, for the worst performing participant the average RMSE was 0.35 BW with an average correlation of 0.979. The inclusion of a participant within hyperparameter optimisation did not appear to considerably influence model performance, with just a 0.01 BW difference between the average RMSE of the two groups.

Figure 3 A box and whisker plot showing the root mean squared error (RMSE) of the Leave One Subject Out (LOSO) validation, using the previously selected hyperparameters for GRFv estimation.

The solid line within each box represents the median error across all foot contacts, the box spans from the 25th to the 75th percentile (interquartile range), and the whiskers encompass 1.5× the interquartile range. The participants included in the hyperparameter optimisation process are shown in the left section of the graph, those not included are shown in the right section.

The mean relative RMSE (rRMSE) (RMSE normalised to the range of the gold standard waveform) across all participants was 3.2%, the mean percentage difference in peak GRFv value was 6.3%, and the mean percentage difference in GRFv impulse was 5.5% (Fig. 4). The majority of participants that the GRFv estimation model achieved low rRMSE values for also had low error in the estimation of peak vertical force values and area under the force trace.

Figure 4 A bar chart showing the relative root mean squared error (rRMSE) (A), percentage difference in impulse (B), and percentage peak difference (C) between the measured and estimated vertical ground reaction force traces.

rRMSE is relative to the range of measured force values, percentage peak difference, and percentage impulse difference is relative to the normalised values. Each bar shows the mean error of all foot contacts for that participant and error bars show ± one standard deviation. The order of participants along the x-axis is sorted from smallest to largest mean GRFv rRMSE values.

Figure 5 shows examples of individual foot contacts across the three gradient stages of the treadmill protocol and across three representative participants. The individual contacts were chosen based on the RMSE of that contact being closest to the mean RMSE for that participant and protocol section. Most notably, it is clear that with the participant that the model performs worst on, the model consistently under estimates the peak GRFv across all conditions.

Figure 5 Example single representative foot contacts for flat (1% gradient), uphill (6% gradient), and downhill (−4% gradient) treadmill running for three example participants.

The left column shows vertical ground reaction forces ( GRFv) from a participant that the model performed best on, the middle column shows data from an average performing participant, and the right column shows data from the worst performing participant. The solid black line shows the force measured from the instrumented treadmill (after 15 Hz cut off filter has been applied), the dashed green line shows the estimated force trace from the deep learning model.

Anteroposterior ground reaction force estimation

The best combination of hyperparmaters for the GRFap model was used to create the final architecture for the GRFap estimation model and LOSO on all 50 participants was performed. Across all participants the average RMSE was 0.04 bodyweight (BW) (Fig. 6) and the average correlation between measured and estimated GRFap traces was 0.982. The accuracy with which the trained model could estimate GRFap appeared to be more consistent than the GRFv estimation results. The same participant, who typically landed very far forward on the forefoot (P24), who had the lowest model performance for both GRFv estimation also had one of the highest levels of error for GRFap estimation. However, it was not consistently the case that those participants that received poorer estimation accuracy from the GRFv model also performed poorly during GRFap estimation.

Figure 6 A box and whisker plot showing the root mean squared error (RMSE) of the Leave One Subject Out (LOSO) validation, using the previously selected hyperparameters for GRFap estimation.

The solid line within each box represents the median error across all foot contacts, the box spans from the 25th to the 75th percentile (interquartile range), and the whiskers encompass 1.5× the interquartile range. The participants included in the hyperparameter optimisation process are shown in the left section of the graph, those not included are shown in the right section.

The mean relative RMSE (rRMSE) across all participants was 3.1%, the mean absolute difference in the peak braking GRFap value was 22.1 N (0.03 BW), and the mean absolute difference in the peak propulsion GRFap value was 19.7 N (0.03 BW) (Fig. 7). It was not the case that as rRMSE increased the error in the estimated peak braking and propulsive forces also increased, which suggests that temporal alignment and estimated curve shape varied between participants. Figure 8 gives examples of how these individual contacts can be constructed. As would be expected the estimated GRFap traces for the participant which the model performed best on showed excellent temporal alignment and shape replication, matched with very close peak braking and propulsive forces during the foot contact. In contrast, the worst performing participant had estimated GRFap traces with poor temporal alignment and atypical signal shape just prior to midstance.

Figure 7 A bar chart showing the relative root mean squared error (rRMSE) (A), difference in peak braking force (B), and difference in peak propulsive force (C) between the measured and estimated anteroposterior ground reaction force traces.

rRMSE is relative to the range of measured force values. Each bar shows the mean error of all foot contacts for that participant and error bars show ± one standard deviation. The order of participants along the x-axis are sorted from smallest to largest mean GRFap rRMSE values.

Figure 8 Example single representative foot contacts for flat (1% gradient), uphill (6% gradient), and downhill (−4% gradient) treadmill running for three example participants.

The left column shows anteroposterior ground reaction forces ( GRFap) from a participant that the model performed best on, the middle column shows data from an average performing participant, and the right column shows data from the worst performing participant. The solid black line shows the force measured from the instrumented treadmill (after 15 Hz cut off filter has been applied), the dashed green line shows the estimated force trace from the deep learning model.

Model sensitivity

Due to the similarity of GRF traces during running, a machine learning model could feasibly tend to output an ‘average’ force trace without much sensitivity to the input values, and still achieve relatively high levels of accuracy. It was seen that during GRFv estimation there was a relatively consistent trend whereby the magnitude of force was overestimated at lower force values and underestimated for larger force values (positive correlation of the line of best fit) (Fig. 9). This finding does indicate that the output of the GRFv estimation models do have a tendency to ‘regress towards the mean’ at the extremities of the training data. This behaviour could be expected within deep learning models due to the loss minimisation approach that is used within their training. However, there are many different techniques that are employed to try and minimise this behaviour. Whilst the identification of this behaviour is important, understanding of the magnitude of it has much greater practical implications. The gradient of the line of best fit across the points presented in Fig. 9 can help quantify this regression magnitude. For example, for the presented model, it could be could be concluded that for every 1 BW increase in peak GRFv above 2.5 BW there would be a 0.3 BW underestimation on average.

Figure 9 A Bland Altman plot that shows the error between the measured and estimated vertical ground reaction force metrics at the range of measured force magnitudes.

Subplot (A) shows difference in the peak force value. Subplot (B) shows the difference in impulse, taken as the area under the force trace, which had been normalised to bodyweight and a length of 400 data points. The solid horizontal lines show the mean error and the dashed lines show the 95% confidence interval area ( ±1.96× Standard Deviations). The individual scatter points indicate whether the foot contact took place in a flat (1% gradient), uphill (+6% gradient), or downhill (−4% gradient) running trial.

The presence of ‘regression towards the mean’ behaviour was less apparent in the estimation of peak GRFap braking force (Fig. 10A), with the error in the estimation of peak braking remaining relatively consistent across the range of magnitudes. However, for those contacts with higher propulsion forces their was a tendency for the model to underestimate their magnitude (Fig. 10B).

Figure 10 A Bland Altman plot that shows the error between the measured and estimated anteroposterior ground reaction force metrics at the range of measured force magnitudes.

Subplot (A) shows difference in the peak braking force value. Subplot (B) shows the difference in the peak propulsive force value. The solid horizontal lines show the mean error and the dashed lines show the 95% confidence interval area ( ±1.96× Standard Deviations). The individual scatter points indicate whether the foot contact took place in a flat (1% gradient), uphill (+6% gradient), or downhill (−4% gradient) running trial.

Discussion

In this study we aimed to use data collected from a consumer-grade wearable system to estimate the ground reaction forces of a large group of runners, with highly varied anthropometric characteristics, whilst they completed treadmill running at many speeds and gradients. Despite the greater variance in the dataset, the input data from the Nurvv Run wearable system combined with a recurrent neural network was able to accurately estimate 2-dimensional ground reaction forces for most participants, matching and even improving on prior results in this research field. The mean rRMSE was 3.2% (range: 0.8–8.7%) for GRFv estimation and 3.1% (range: 1.3–17.3%) for GRFap estimation. An additional notable finding in the study is the drop off in the GRF estimation accuracy for certain runners. Those individuals were often runners with a more predominant forefoot strike. This often overlooked understanding of the model’s current limitations is essential for the eventual deployment of remote data collection methods, and should inform the interpretation of the data collected.

The force estimation accuracy achieved in this article is in line and even improves on some of the previous studies that aimed to estimate GRFs during running using machine learning and wearable sensors. Alcantara et al. (2021) collected data from three biaxial accelerometers amongst 19 runners that completed a treadmill protocol of varying speeds and inclines, training an LSTM model to estimate GRFv. Using a similar LOSO evaluation method to that within this article, they found that rRMSE ranged from 5.4–7.3% across participants. In research by Honert et al. (2022), a similar treadmill protocol was completed but wearable data was instead collected from a research grade pressure sensor. An LSTM model architecture was again employed and both a GRFv and GRFap trace were estimated for each contact. Their results show a rRMSE range across participants of 4.0–8.6% for GRFv estimation and 6.4–15.4% for GRFap estimation. Similar studies have also previously been completed during level overground running, both Johnson et al. (2021) and Dorschky et al. (2020) evaluated the ability of CNNs to estimate GRF traces utilising data from multiple IMUs. The lower errors were reported by Dorschky et al. (2020), achieving a rRMSE 6.1–9.1% for GRFv estimation and 5.5–8.9% for GRFap estimation.

Across the many recent studies that have aimed to estimate kinetic time series signals there have been a wide array of metrics used to try and communicate the accuracy of the model’s output. The R2 value between the estimated and measured signals (Johnson et al., 2021), the RMSE between the two signals (Kim et al., 2023), RMSE normalised to body weight (Ancillao et al., 2018), RMSE normalised to body mass (Cerfoglio et al., 2021), statistical comparison between the two signals using Statistical Parametric Mapping (SPM) (Honert et al., 2022) to name a few. This lack of consistency makes comparison between studies challenging and limits the ability to identify the more effective methods. Another point of variation is the inclusion of flight phases within the signal comparison, where as long as the model being used can correctly identify this phase the similarity of the signal will be artificially inflated. Another factor to consider when comparing between studies is the model evaluation method, the use of LOSO in this study allowed the identification of a small number of participants for which the model performed considerably worse. If an alternative method such as cross fold validation would have been used the errors from these individuals could have been masked and greater confidence in the model’s capability could have been concluded.

The identification of these outlying performances prompted further investigation into the cause of the elevated errors amongst certain participants. Figure 11 shows the association between the anteroposterior CoP position at foot touchdown and GRFv estimation accuracy. It can be seen that a large increase in estimation error occurs for participants landing much further forward on the foot. It could be hypothesised that the lower volume of training data within this region led to the higher levels of estimation error. However, a similarly low level of training data is shown in the midfoot region without a large elevation in GRFv estimation error for those contacts. With the foot contacts only occurring in the forefoot region, the entirety of the foot’s contact with the floor is being evaluated by a smaller proportion of the 16 FSR sensors in the insole, effectively reducing the resolution of information about the foot contact being fed to the model. Secondly, the association between the force applied to the FSR sensor and its raw output value is logarithmic. This means that at higher force values a given change in the magnitude of the force being applied causes a much smaller increase in FSR output value, which also works to reduce the resolution of the measure. With forefoot contacts having the force split amongst a much fewer number of sensors, it could be hypothesised that the FSRs are more likely to be operating within this lower resolution range.

Figure 11 A graph showing how mean error between the measured and estimated peak GRFv varied with different center of pressure (CoP) locations at touchdown.

Subplot (A) shows the quantity of foot contacts in the dataset at each touchdown position. Subplot (B) shows the average error (black line) in the GRFv peak that was output from the model with respect to each touchdown position. The shaded red region shows ± one standard deviation.

Figure 12 shows the PFI score for each input variable, the data encapsulated by the box plot is the median PFI score across all foot contacts used during training for each participant. The most noticeable element of these results is the large variation in the ‘importance’ of certain variables amongst participants, meaning that for certain participants, swapping out an input variable vector with a sequence of random numbers drastically reduced the model performance but for other participants this change had minimal effect on model performance. There are several possible interpretations of this variation, one is that based on subject and task specific variations certain inputs have more or less relevance to accurate estimation of force. For example, as force traces were all normalised to 400 data points it could be that for those contacts that had a ground contact time (GCT) close to the chosen normalisation length of 400, the ‘loss’ of this information was not highly influential to the output. Whereas, for those contacts with a duration much further from the normalised length, the loss of this information becomes much more relevant to the model’s capability of estimating the force trace accurately.

Figure 12 A box and whisker plot showing the Permutation Feature Importance (PFI) value for each input variable, shown for both vertical (blue) and anteroposterior (red) ground reaction force estimation models.

The solid line within each box represents the median PFI value across all participants, the box spans from the 25th to the 75th percentile (interquartile range), and the whiskers encompass 1.5× the interquartile range. Participants with an average PFI outside of this range were shown as a scatter point.

The similarity of PFI values and their proximity to 1.0 across the 16 pressure channels could be interpreted as there being high levels of redundancy between these inputs. Meaning that the manipulation of just one of these channels, whilst keeping all others the same, did not significantly influence model performance, perhaps due to the same information being encoded by data from the other time series channels. Two of the most clear elevated mean PFI magnitudes were that of the anteroposterior acceleration signal from the IMU (1.56) and the treadmill incline value (1.52) during GRFap estimation, confirming their expected relevance for estimating GRFap shape and magnitude.

There are certain limitations within this study that should be acknowledged with a view to its intended application. Firstly, due to the presence of low frequency vibrational noise within the instrumented treadmill force data collected at an incline, the signal had to be heavily filtered, as has been done in previous studies (Matijevich et al., 2019). Whilst this is a required step it is likely that some of the ‘true’ force signal would have been lost on top of removing the vibrational noise. The vast majority of a runner’s training typically occurs during overground running on a variety of different surfaces rather than on a treadmill. Future research should look to evaluate how the developed models perform on alternative running surfaces and whether models trained on a relatively large volume of treadmill data can be used as a starting point for future GRF estimation models intended for varied running surfaces.

Whilst the greater variance captured by the dataset collected in this study would have worked to minmise ‘regression towards the mean’ within the model output, it also allows for a greater potential to identify such behaviour, in comparison to a more homogeneous dataset. Typical regularisation approaches such as dropout layers were implemented to try and further minimise regression towards the mean behaviour. However, alternative deep learning architectures could also be evaluated in an attempt to improve upon the model performance presented in this study. Transformer models, in particular, leverage ‘attention layers’ that facilitate dynamic adjustments to the model’s weights during inference, allowing it to adapt to the input data’s distinctive characteristics (Vaswani et al., 2017). These adjustments could help make the model more robust to differing running techniques or anthropometric characteristics. An alternative approach that is beginning to become more commonly used within this space is the use of physics informed deep learning models (Taneja et al., 2022). Rather than learning the associations between input and output data in isolation, these approaches can help ground the model in physics-based constraints. If implemented correctly then the use of these constraints could also make the model more robust to input data that was either less commonly seen within the training dataset or outside its bounds. Within this context, that could mean mean more accurate GRF estimation for runners with technique characteristics less commonly seen, such as landing very anterior on the foot.

Finally, collecting data from a larger pool of runners should eliminate the amount of runners with ‘atypical’ running technique characteristics. The model training would then be occurring with a more balanced and representative set of runners, reducing the requirement for the model to infer input-output relationships outside the bounds of the training set. Previous studies within this space have also shown that the augmentation of collected datasets with synthetically created data can improve GRF estimation (Eguchi & Takahashi, 2019).

Conclusion

This study evaluated the ability of an LSTM deep learning model fed with wearable data from a consumer grade sensor system to estimate vertical and anteroposterior ground reaction forces, amongst a highly varied pool of runners and running conditions. The mean rRMSE across all participants and running conditions was 3.2% and 3.1% for GRFv and GRFap estimation respectively. The findings showcased the ability for the model to sensitively estimate variations in the force trace based solely on a foot-based consumer-priced wearable system. Future research should look for further improvements in the models’ robustness to unique running characteristics, such as extreme forefoot contacts, and evaluate its performance during overground running. Upon completion, this approach could be used to effectively collect GRF data outside of laboratory environments for a much larger percentage of a runner’s training exposure. Such a tool could be used to significantly scale-up injury risk factor studies for running and provide runners with more accessible performance feedback.

Additional Information and Declarations

Competing Interests

Author Contributions

Human Ethics

Data Availability

Author Grant Trewartha was a former employee at Nurvv Ltd. when this project started. Nurvv Ltd. has since been dissolved.

Josh Carter conceived and designed the experiments, performed the experiments, analyzed the data, prepared figures and/or tables, authored or reviewed drafts of the article, and approved the final draft.

Xi Chen conceived and designed the experiments, authored or reviewed drafts of the article, and approved the final draft.

Dario Cazzola conceived and designed the experiments, authored or reviewed drafts of the article, and approved the final draft.

Grant Trewartha conceived and designed the experiments, authored or reviewed drafts of the article, and approved the final draft.

Ezio Preatoni conceived and designed the experiments, authored or reviewed drafts of the article, and approved the final draft.

The following information was supplied relating to ethical approvals (i.e., approving body and any reference numbers):

The University of Bath Research Ethics Approval Committee for Health reviewed this study (reference number: EP 20/21 069).

The following information was supplied regarding data availability:

Example code to read in the npy files and train LSTM models to predict chosen GRF variables is available at GitHub:

https://github.com/JoshCarter97/LSTM_GRF_Estimation.

Data, including an .npy file for the force data, pressure data, and IMU data for each usable trial, are available at OSF:

Carter, Joshua A. 2024. “Consumer-Priced Wearable Sensors Combined with Deep Learning Can Be Used to Accurately Predict Ground Reaction Forces during Various Treadmill Running Conditions.” OSF. June 26. osf.io/wqjp9.

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
