# Peer review of "Consumer-priced wearable sensors combined with deep learning can be used to accurately predict ground reaction forces during various treadmill running conditions"

_PeerJ, doi:10.7717/peerj.17896_

## Round 0.1 · original submission · Major Revisions

Thank you for submitting your manuscript, “Consumer-priced wearable sensors combined with deep learning can be used to accurately predict ground reaction forces during various treadmill running conditions” for consideration in PeerJ. We have received evaluations from two external reviewers. Though Reviewer 1 recommends Minor Revisions, Reviewer 2 had more substantial concerns, recommending Major Revisions. Having read the manuscript, I concur with the suggested edits of both reviewers and urge you to take a detailed look at their questions/concerns. Following from this, I am recommending Major Revisions.

In addition to the concerns raised by the reviewers, I had one minor comment: I don’t think relative RMSE is ever defined (i.e., relative to what – the empirical value?). This may be a common metric in this literature, but it would be good to explicitly define for the reader.

Reviewer 1 ·

Basic reporting

This manuscript is clearly written, has an appropriate background and introduction, establishes clear hypotheses that will be tested, and describes how this study fits in the existing literature. The language is professional and appropriate and the figures are well designed and clear. Raw data and code are provided via online repositories. The only issue here is that there appear to be formatting issues for citations throughout the text. The authors cite using Author (YEAR) within the text when the name of the author(s) is not intended to be stated in the sentence. There are also many instances where citations are repeated (e.g., Lines 73-74 where it says Johnson et al. (2021) Johnson et al. (2021).

Experimental design

The research question is well defined and supported by appropriate data. The authors clearly define their methods and provide appropriate and thorough analysis of their findings and explore sources of error and variation through a variety of methods and are supported by well-designed figures. Overall, this study is well-executed and the authors have designed a clever way to integrate data from multiple sources. I cannot evaluate the use of deep learning models as this is not my area of expertise, but the methods seem reasonable and are consistent with those used in the literature.

Validity of the findings

The results from this study appear to be valid and add to the literature. All raw data and code to replicate the analyses appear to be provided. The results appear to be statistically sound and the authors have provided appropriate analyses of error and variation, particularly where the model doesn't fit the raw data. The results match the research question and are appropriate and interesting. The conclusions are well stated and the authors provide an appropriate statement of limitations of this study and future directions.

Additional comments

[Line 68] You should define GRFv
[Line 73] You should define RMSE at this point since you haven’t defined it yet
[Line 77] Here, you actually write out ‘root mean squared error’ and could just use RMSE
[Line 123] Why was the 1000 Hz data down-sampled for storage limitations then up-sampled back to 1000 Hz? Was the storage limitation on the Nurvv device?
[Line 164] You should define GRFap on first instance

Do you have any hypotheses for why certain individuals had poor fits for vGRF but not GRFap? The authors provided plausible explanations for poor vGRF fits, but not GRFap. The authors did point out elevated PFI values for AP acceleration from the IMU and treadmill incline, but I'm not sure if this was linked to the overall poor fits of GRFap in some individuals.

Reviewer 2 ·

Basic reporting

- I found this paper to be clearly written in professional English.

- There are several instances where citations appear to be repeated (Intro paragraph 4: Honert et al. 2022 and Johnson et al. 2021 are repeated; again in the Discussion section). Perhaps this is a formatting or LaTeX importing quirk?

- Many of the figures (Figs 3, 4, 6, 7, 12) have a font size that is difficult to read, particularly for the subject numbers (x-axes). To help with readability, I would recommend sizing the figures such that the smallest font is approximately the same size as the font size in the caption.

- It may help to label sub-figures as (A) and (B).

- Are Figures 5 and 8 plotting a single step from representative participants, or averaged steps? If the latter, please include error bars or standard deviation bars (akin to Fig 11B) so that the reader can appropriately contextualize the accuracy of the model.

Experimental design

- This work fits the aims and scope of the journal.

- The specific research question of this work could be more clearly articulated. My interpretation is that the primary contribution of this work is to use an inexpensive, consumer-grade wearable insole and IMU set to predict 2D GRFs using a slightly improved version of a previously-validated machine learning model in healthy runners across a range of running speeds. If the research question is to ask whether consumer-grade devices can attain similar accuracies as laboratory-grade devices, I would expect a comparison of the two. At minimum, a price comparison between the device used in this study and that used by (for instance) Alcantara and colleagues would help to make these results more meaningful. At present, I am having a difficult time understanding the knowledge gap addressed in this work.

- An additional aim of this study seems to be the goal of assessing the model's fit on runners of varied anthropometric characteristics. There was a brief interpretation of how a single subject's forefoot striking contributed to a high error, but I would have liked to see a deeper engagement with this explanation, in particular to back up the statement in the Discussion that the runners with the highest error "were often runners with a more predominant forefoot strike." For example, in Table 1, the percentage of forefoot-striking runners is listed; what were the respective errors for these three groups? To that end, how was striking pattern defined? I apologize if I missed the explanation somewhere.

- To add to the previous point -- might the authors consider looking at whether all of the subject characteristics outlined in Table 1 have an impact on the accuracy of the model? For example, the PFI values for Body Mass make it seem that it played a large role. As an aside, details on PFI should be added to the Methods section rather than reported for the first time in the Discussion.

- The authors mention that the models "do have a tendency to 'regress towards the mean' at the extremities of the training data." This is an important limitation and I am glad to see it addressed. The authors then state that there are "many different techniques that are employed to try and minimise this behaviour" and that "understanding the magnitude of it has much greater practical implications." Both points are excellent! I was then very disappointed to see the paragraph end here. What are the techniques that could be employed here to minimize regression to the mean, and what reasons do the authors have for not implementing them in this model? What is the magnitude of this error, and how significant is it in, for instance, a clinical context? How would an error of 0.4 BW in estimating the vGRF impact a runner's knowledge of their risk for injury? As this paragraph is in the Results section, I was expecting more quantitative results; otherwise, these sentences may be better suited for the Limitations section in the Discussion.

- "If they were able, participants also completed 3 minutes at the fixed speed of [...]" -- how many participants did/did not complete this portion of the study?

- The IMU was placed on only the left foot, but in Data Processing it is mentioned that COPx and COPy were calculated from the "pressure data from each of the 16 FSR channels under each foot." I was confused here -- could you clarify whether you were predicting GRFs under both feet, or only the left foot?

- A key limitation of this work that stands in the way of the authors' goal of "provid[ing] runners with real-time performance feedback" is that the system does not work in real-time, and requires significant pre-processing, temporal alignment, and normalization. I would appreciate a deeper engagement with this limitation in the Discussion. Is this limitation easily addressed?

Validity of the findings

- I was pleased to see the data and code provided. When examining one of the scripts (Run_LOSO.py) I noticed that the hyperparameters set for the vGRF model were not the same as those reported in the paper (lines 38-45). It's a small suggestion, but to make it easy for readers to replicate your results, I might consider setting these to be consistent with those reported.

- I couldn't find a place to set the random seed to a fixed value, so due to the inherent noise in your ML model, readers will not be able to replicate the exact values reported here. I would consider making your trained model public as well -- this will also help in the translation of your findings, as readers without access to the necessary hardware will be able to use the trained model on their own data.

- In the final section of the Discussion, the authors write that modifying the model to use physics-informed constraints could help the model to predict GRFs outside of the training set. I am not certain that a PINN would be beneficial in this scenario, as forces from the insoles would be known -- the neural network component of the PINN seeks to minimize the residuals of the PDEs describing the movement, and would overfit to the measured GRFs. This is not my primary area of expertise, however, and an interested reader might appreciate more reflection from the authors on what and how "physics-based constraints, if implemented correctly" would help to prevent regression to the mean.

Additional comments

In summary, I enjoyed reading this manuscript and believe that it deserves a round of thoughtful reflection in order to become its best possible contribution. Thank you!

---

## Round 0.2 · accepted · Accept

Both reviewers concur that the revisions to this manuscript thoroughly address all prior concerns. I'm happy to recommend acceptance.

Reviewer 1 ·

Basic reporting

This manuscript is clearly written, has an appropriate background and introduction, establishes clear hypotheses that will be tested, and describes how this study fits in the existing literature. The language is professional and appropriate and the figures are well designed and clear. Raw data and code are provided via online repositories.

Experimental design

The research question is well defined and supported by appropriate data. The authors clearly define their methods and provide appropriate and thorough analysis of their findings and explore sources of error and variation through a variety of methods and are supported by well-designed figures. Overall, this study is well-executed and the authors have designed a clever way to integrate data from multiple sources. I cannot evaluate the use of deep learning models as this is not my area of expertise, but the methods seem reasonable and are consistent with those used in the literature.

Validity of the findings

The results from this study appear to be valid and add to the literature. All raw data and code to replicate the analyses appear to be provided. The results appear to be statistically sound and the authors have provided appropriate analyses of error and variation, particularly where the model doesn't fit the raw data. The results match the research question and are appropriate and interesting. The conclusions are well stated and the authors provide an appropriate statement of limitations of this study and future directions.

Additional comments

I believe the authors have addressed reviewer concerns and the manuscript is publishable in its current state.

Reviewer 2 ·

Basic reporting

no comment

Experimental design

no comment

Validity of the findings

no comment

Additional comments

Thank you for taking the time to read through the feedback and thoughtfully address all points raised. I have no further comments at this time.